

# Seasonal monitoring of melt and accumulation within the deep percolation zone of the Greenland Ice Sheet and comparison with simulations of regional climate modeling

Heilig Achim[1], Eisen Olaf[2,3], MacFerrin Michael[4], Tedesco Marco[5,6], and Fettweis Xavier[7]

[1]Department of Earth and Environmental Sciences, LMU, Munich, Germany
[2]Alfred Wegener Institute Helmholtz-Centre for Polar and Marine Research, Bremerhaven, Germany
[3]Department of Geosciences, University of Bremen, Bremen, Germany
[4]Cooperative Institute for Research in Environmental Sciences, University of Colorado, Boulder, CO USA
[5]Lamont-Doherty Earth Observatory, Columbia University
[6]NASA Goddard Institute of Space Studies, New York, USA
[7]Department of Geography, University of Liège, Liège, Belgium

*Correspondence to:* Achim Heilig (heilig@r-hm.de)

**Abstract.** Increasing melt over the Greenland ice sheet (GrIS) recorded over the past years has resulted in significant changes of the percolation regime of the ice sheet. It remains unclear whether Greenland's percolation zone will act as meltwater buffer in the near future through gradually filling all pore space or if near-surface refreezing causes the formation of impermeable layers, which provoke lateral runoff. Homogeneous ice layers within perennial firn, as well as near-surface ice layers of several

meter thickness are observable in firn cores. Because firn coring is a destructive method, deriving stratigraphic changes in firn and allocation of summer melt events is challenging. To overcome this deficit and provide continuous data for model evaluations on snow and firn density, temporal changes in liquid water content and depths of water infiltration, we installed an upward-looking radar system (upGPR) 3.4 m below the snow surface in May 2016 close to Camp Raven (66.4779°N/ 46.2856°W) at 2120 m a.s.l.. The radar is capable to monitor quasi-continuously changes in snow and firn stratigraphy, which

occur above the antennas. For summer 2016, we observed four major melt events, which routed liquid water into various depths beneath the surface. The last event in mid-August resulted in the deepest percolation down to about 2.3 m beneath the surface. Comparisons with simulations from the regional climate model MAR are in very good agreement in terms of seasonal changes in accumulation and timing of onset of melt. However, neither bulk density of near-surface layers nor the amounts of liquid water and percolation depths predicted by MAR correspond with upGPR data. Radar data and records of a nearby thermistor string, in contrast, matched very well, for both, timing and depth of temperature changes and observed water percolations.

All four melt events transferred a cumulative mass of 56 kg/m$^2$ into firn beneath the summer surface of 2015. We find that continuous observations of liquid water content, percolation depths and rates for the seasonal mass fluxes are sufficiently accurate to provide valuable information for validation of model approaches and help to develop a better understanding of liquid water retention and percolation in perennial firn.





*Copyright statement.* TEXT

# 1 Introduction

The Greenland ice sheet (GrIS) has been affected by changes in environmental conditions over recent decades, which resulted in an average negative mass balance all over the ice sheet (e.g., Sasgen et al., 2012). Mass loss of the ice sheet, determined by methods relying on satellite data, multiplied by a factor of four within the last two decades, from 51 ± 65 Gt per year (1992 – 2001) to 211 ± 37 Gt per year in 2002 – 2011 (Shepherd et al., 2012; Hanna et al., 2013). Negative annual surface mass balances over the same time period are attributed to an increase in surface melt and runoff (Vaughan et al., 2013). The recent mass loss is ascribed to 61% to a decrease in surface mass balance and to 39% to an increase in solid ice discharge (van den Broeke et al., 2016). Since melt conditions are expected to continue (Vizcaíno et al., 2010; Huybrechts et al., 2011) and being amplified especially in northern latitudes (e.g., Meehl et al., 2012), the determination of melt and refreezing, and mass redistribution through liquid water are of utmost importance for density and firn temperature estimations in accumulation areas of polar regions (e.g., Gascon et al., 2014). Moreover, increased surface melt influences entire glacier systems including glacier velocities and basal motion (e.g., Meierbachtol et al., 2013). Single snow and firn parameters such as density and temperature have a major effect on the storage capacity of melt water with the consequence that understanding and monitoring of these parameters is necessary for correct predictions of surface mass balance and, hence, on sea-level rise through melt of polar ice sheets (e.g., Hanna et al., 2008; Gardner et al., 2013). Liquid water infiltration into snow and firn and retention therein cause a large fraction of uncertainty in current surface mass balance measurements and projections (Vernon et al., 2013), because observations are lacking (Harper et al., 2012).

Recently, near-surface firn layers (upper tens of meters) are exposed to enhanced effects from mass loss, firn compaction and refreezing. Although records for the maximum extent in area of surficial melt on the GrIS were broken in 2005 (Hanna et al., 2008), 2007 (Tedesco et al., 2008), 2010 (Tedesco et al., 2011) and 2012 (e.g., Tedesco et al., 2013), for none of these record years direct determinations in firn of percolation depths and quantification of the amount of melt are available. In situ data usually just exist for the area extent of surficial melt over the GrIS (e.g., Abdalati and Steffen, 2001).

For percolation regimes, it remains unclear whether meltwater is stored and refreezes within the firnpack and gradually fills up all pore space or whether near-surface refreezing causes the formation of impermeable ice slabs (MacFerrin et al., submitted). Such thick ice lenses block water infiltration and thus force lateral runoff. Both, homogeneous ice layers within perennial firn (Harper et al., 2012), as well as near-surface ice layers of several meter thickness are observable in firn cores (Machguth et al., 2016). However, the formation process of neither of them in real time has been observable before. Machguth et al. (2016) state that it is inevitable to understand feedback mechanisms in firn to predict future GrIS mass balances. Taking firn cores is a destructive sampling technique and thus hampers monitoring and derivation of quantification of changes in parameters. It remains nondistinctive whether differences in between annual cores are attributed to spatial variability or temporal evolution.

Coverage of in situ observations in space and time is insufficient to produce detailed maps for seasonal mass balance (van den Broeke et al., 2017). Hence, regional climate models are used to reproduce the contemporary and previous surface mass





balances (Fettweis et al., 2017; Noël et al., 2017) and to predict future mass changes. Apart from few existing automatic weather stations (AWS), no temporal continuous observations exist to validate results of such models. AWS, however, provide only limited information about changes in snow- and firnpack parameters. No direct data for percolation, snow and firn density and mass transfers are available from atmospheric data. Data on refreezing within snow and firn can only be derived indirectly

from temperature data (Steger et al., 2017a). However, the quantification of surface water in combination with accumulation and monitoring of liquid water percolation and blocking capabilities of ice layers has been defined as very valuable by an expert elicitation and recent model intercomparison (e.g., van As et al., 2016; Steger et al., 2017b). Temperature records in snow and firn (e.g., Humphrey et al., 2012) only indicate the depth of percolating meltwater but cannot provide information on mass fluxes and bulk liquid water content.

Upward-looking ground penetrating radar systems (upGPR) (Heilig et al., 2009, 2010) proved to provide reliable data on bulk snow height and density, liquid water infiltration, volumetric liquid water content ($\theta_w$) as well as total accumulation (SWE) in seasonal snow (Mitterer et al., 2011; Schmid et al., 2014; Heilig et al., 2015). For this study, we installed an upGPR in perennial firn within the deep percolation zone of the GrIS. Such instrumentation is capable of providing new insights in the temporal evolution of ice layer formation, liquid water percolation and of monitoring differences in summer melt for

various melt seasons. On a longer term perspective, upGPR might be capable of monitoring processes and changes which lead to establishment of either impermeable ice slabs or the progressive fill-up of pore space above the system. To estimate the reliability of radar-derived parameters, we compare determined percolation depths with changes in temperature records and analyze monitored increases in SWE with results of the regional climate model MAR (e.g., Fettweis et al., 2017) for the closest grid point. In addition, to validate performance of MAR at a point scale, we investigate discrepancies in near surface densities,

percolation depths and bulk liquid water content between simulations and radar data. The presented data have a large potential to demonstrate current short comings in model approaches and supports understanding of short-term changes in snow and firn of near-surface layers. Such data will help to improve understanding of liquid water retention by quantification of surface water in combination with accumulation and monitoring liquid water percolation and blocking capabilities of ice layers.

## 2 Methodology

### 2.1 Test site, instrumentation and data processing

We installed an upGPR system within the perennial firn regime of the GrIS at the research site Dye-2 (Coordinates: 66.4779°N/ 46.2856°W) next to Camp Raven, in April 2016 (Figure 1). The radar system consists of an IDS (Ingegneria dei Sistemi, Pisa, Italy) FastWave control unit with dual frequency 600/1600 MHz antennas. The whole aperture is powered by six 50 Ah batteries and two 60 W solar panels (Figure 1b). We buried the radar antennas in a box at approximately 4.5 m beneath the surface of

April 2016. To enable observation of undisturbed snow and firn, we further excavated an additional 2 m cave sideways and fixed the antenna box at this position. The upGPR system is programmed to conduct measurements periodically at three different intervals: during summer time (15 April – 14 August) every 30 min during the day (9:00 h –21:00 h) and every 1 h for nocturnal measurements; after 14 August until 14 October and from 01 March until 14 April every 3 h and from 15 October until end



of February, we only record one measurement per day at 12:00 h. All times are given in local winter time (UTC –3 h). From 16 October 2016 on until our next visit in April 2017, the radar stopped working due to technical problems. For analysis, we defined the start of upGPR measurements to 1 May 2016, when the installation pit was filled in and had had time to settle for 2 days.

Radar data were processed as described in Schmid et al. (2014). Snow surfaces in the resulting radargrams for both frequencies were determined using the "semi-automated picking algorithm" (Schmid et al., 2014). All reflectors were automatically picked at the maximum amplitude per positive half cycle or minimum amplitude per negative half cycle, depending on the phase sequence of the respective reflector. However, for the same reflector, we consistently chose the same half cycle. The resulting radargram of the 1600 MHz system was used to pick the snow surface and the 600 MHz signal to determine the two-

way travel time (TWT with mathematical symbol $\tau$) to the target reflector. However, for periods with large amounts of melt affecting the snow- and firnpack, the reflection from the snow surface for the 1600 MHz antennas diminished. We then also used the 600 MHz signal to pick surfaces for such periods and vice versa used higher frequency signal to determine the distance of the target reflector for some radar records. For all displayed radargrams, we generated a velocity model for electromagnetic waves derived from core densities to convert measured TWT to height above the upGPR antennas. Since we only have density

data available for May when we visited the site, the velocity model is not updated during the season and certainly incorrect for radar reflections affected by liquid water. These inaccuracies have no influence on data analysis as will be shown in the discussion. The model is just used for visualization.

Two firn cores down to the depth of the radar antennas were drilled in 2016 and used for the installation of the target reflector (Figure 1). Core data were processed in 5 cm steps for average densities and stratigraphy was visually inspected on

a 1 cm resolution. In May 2017, we drilled only one core down to 5.5 m depth in close proximity of the radar antennas but outside of the estimated footprint of the antennas (about 8 m away from the center of the antennas). Again data were processed with 5 cm resolution in density and 1 cm resolution in stratigraphy.

In 2016 in addition to the radar, we also installed a thermistor string about 4 m apart from the solar panel mast of the radar system (Figure 1b). Thermistors were deployed at depths of 0.4 m to 5.4 m at each meter and in addition at 7.4 m and at 9.4 m

depth beneath the snow surface of 1 May 2016.

## 2.2   Determination of bulk snow and firn parameters above the radar antennas

The bulk layer ($L_s$) above the antennas (Figure 1) has a layer thickness $\overline{L_s} = \Sigma \overline{L_i}$, with $i$ the individual layer from one horizon to the next above. Correspondingly, the bulk mass ($b_s$) is the sum of mass of all layers: $b_s = \Sigma b_i$. To derive snow and firn parameters for $L_i$, we use the target reflector at a fixed height above the surface similar to Heilig et al. (2015). With the known

distance between target and antennas ($d$), the surface pick in measured TWT and the known relative dielectric permittivity of air ($\varepsilon_a$), we can simply calculate for the height of the the target above the snow surface ($d_A$). Since the target posts are drilled to the same depth as the radar antennas (Figure 1a), we expect compaction of the radar and the target to be approximately equal. As a consequence, $d$ remains constant. From simple subtraction, we obtain the bulk thickness of the snow and firn layer above the antennas $\overline{L_s} = d - d_A$. The retrieval of bulk firnpack parameters above the antennas relies on previously published





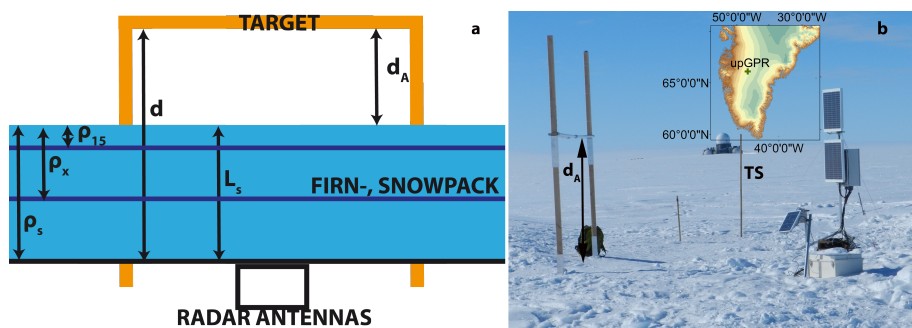

**Figure 1.** Sketch and image of the radar arrangement for Dye 2. (a) Sketch of the installations above and beneath the snow surface. $\rho_i$ indicate the density of specific snow layers, $L_s$ indicates the whole firn and snow column above the antennas and $d_A$, $d$ indicate distances. (b) Image of the above snow installations at the research site Dye-2. The inset displays the location of the upGPR for the southern half of Greenland. The color coding for the inset map represents 250 m contour lines with the digital elevation model generated from Howat et al. (2014). TS in (b) represents the location of the thermistor string.

assumptions and equations (Heilig et al., 2009, 2010; Schmid et al., 2014; Heilig et al., 2015): we used the three phase mixing formula postulated by e.g. Roth et al. (1990) or Wilhelms (2005) with the exponent $\beta = 0.5$ and the assumption of only three contributing volume fractions (air, ice and water: $\theta_a + \theta_i + \theta_w = 1$). For cold conditions with snow and firn temperatures below $0°$ C ($\theta_w = 0$), the bulk density above the antennas can easily be determined. In contrast to conditions in seasonal snow

described by Heilig et al. (2015), melting snow and firn on cold ice sheets can rapidly refreeze due to the underlying cold content. As a consequence, the assumption of a constant ice volume fraction after initial melt is invalid for ice sheets. Hence, melt and dry periods have to be treated differently. The resolution of the thermistor string with a 1 m spacing and the first thermistor at 0.4 m depth is not adequate to identify first occurrences of melt above the antennas. We use radar data instead for identification of timing of melt periods. Surficial melt produces strong changes in dielectric permittivity and, consequently, has

an effect on radar response. The appearances of multiples or ringing in the radargram above the snow surface indicate those effects (Figure 3). This allowed for the determination of periods when melt is present. For such periods, we assume that (i) no lateral flow transported mass downslope (slope angle below $0.5°$); (ii) wind erosion after surficial wetting is not effective; (iii) evaporation and sublimation effects are negligible for wet snow surfaces; (iv) no mass transfer from $L_s$ to layers below is possible as long as percolation did not reach the location of the antennas. Those four assumptions lead to the fact that a

decrease in height of $L_s$ is compensated by a corresponding increase in wet snow density ($\rho_s$), since the total mass ($b_s$) cannot diminish:

$$b_s = \overline{L_s}\rho_s. \tag{1}$$



To reduce the effects of single outliers and uncertainties in the surface and target picks, we averaged the 37 radar measurements per day and analyzed subsequently for diurnal differences during melting periods. For calculating diurnal changes in $\rho_s$, we use equation (1) and determine the differences $(\Delta \overline{L_s})$ from day $i$ to $i+1$ in $\overline{L_s}$:

$$
\rho_{s,i+1} =
\begin{cases}
\overline{L_{s,i}}\rho_{s,i} + \Delta\overline{L_s}\rho_n & if\ \Delta\overline{L_s} < 0 \\
\overline{L_{s,i}}\rho_{s,i}\overline{L_{s,i+1}}^{-1} & if\ \Delta L_s \geq 0
\end{cases},
\tag{2}
$$

with the new snow density estimate $\rho_n = 120\,\text{kg/m}^3$ being slightly larger than for Alpine sites (Schmid et al., 2014).

In a second step, we set the obtained average values per day of $\rho_s$ to be equal for each diurnal radar measurement. Since $\rho_s$ in equation (2) describes the wet snow density, it is impossible to discriminate for individual volume fractions. Hence, we use the empirical equation by Denoth (1994):

$$
\varepsilon_s = 1 + c_1\rho_s + c_s\rho_s^2 + c_3\theta_w + c_4\theta_w^2,
\tag{3}
$$

with $c_1 = 1.92\times10^{-3}$, $c_2 = 4.4\times10^{-7}$, $c_3 = 18.7$, $c_4 = 45$, $\rho_s$ with units [kg/m$^3$] and $\varepsilon_s$ as the relative dielectric permittivity of snow to solve for $\theta_w$.

We checked the reliability of the application of, first, the three phase mixing formulation to gather snow density from defined relative dielectric permittivity ranges for snow and ice ($\varepsilon_s = [1:3.2]$ in increments of 0.01) and applied the received values in equation (3). In case the three phase mixing formulation and the empirically determined equations were compatible, we would receive a volumetric liquid water content of constantly $\theta_w = 0$. Figure 2 displays the estimated discrepancy in $\theta_w$ values. In order to correct for the observed discrepancies, we applied a quadratic correction on the resulting $\theta_w$ of equation (3):
$\theta_{wc} = \theta_w - 1.55 \times 10^{-8}\rho_s^2 + 1.13 \times 10^{-5}\rho_s + 4.10 \times 10^{-6}$ (again with $\rho_s$ in [kg/m$^3$]).

## 2.3 Seasonal mass fluxes

Mass fluxes from snow above the previous summer horizon into firn are hereinafter defined as seasonal mass fluxes (SMF with mathematical symbol $F$). Determination of SMF require more iterations but can be accomplished with the applied setup as well. Two more layer definitions were necessary to prepare SMF analysis. First, we had to define a reference horizon, below which no temporal changes in stratigraphy are observable (Figure 3, yellow line). Consequently, the total mass of the layer between the top of the antennas and the reference horizon did not change within the observation period. From the known height of the reference horizon and corresponding layer thickness $(\overline{L_{s,x}})$, determined from core data, and the calculated $\rho_{s,x}$, we could then continuously calculate the amount of mass of the reference layer $(b_x)$, which results in:

$$
b_x = \overline{L_{s,x}}\rho_{s,x}.
\tag{4}
$$

The second horizon necessary to determine SMF is the previous summer surface. The assignment of the 2015 summer horizon is possible for both radar frequencies over the entire observation period (Figure 3, white line). We chose to refer





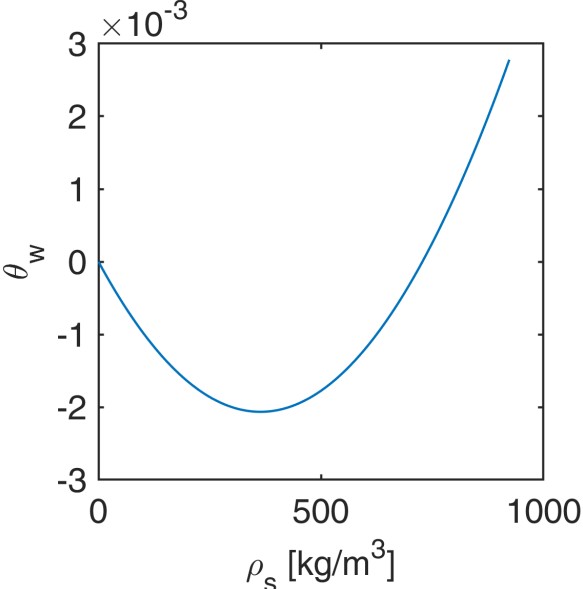

**Figure 2.** Discrepancy of calculated $\theta_w$ from the three-phase mixing formula with exponent $\beta = 0.5$ and Equation (3) for given snow densities ($\rho_s$) and dry snow dielectric permittivities.

to the 600 MHz data (Figure 3, b), since, both, the surface reflection and the summer 2015 horizon are more predominant and persistent for this antenna configuration. It is possible that either the data processing or slightly different environmental conditions influenced radar acquisitions with the consequence that peaks in amplitude shifted by $\pm 1$ sample. During dry snow periods when no compaction of the layer between the reference horizon and the summer 2015 horizon ($L_{15,x}$) was identifiable,

we used the most frequently occurring TWT for both horizons to minimize effects of shifted peaks. To calculate the mass changes ($b_{15}$) occurring within the snow layer above the previous summer surface ($L_{15}$), we had to determine the mass flux ($F_{15,x}$) into $L_{15,x}$ due to percolating melt water. To solve for $b_{15}$, we simply subtracted $b_{15,x}$ together with the seasonal mass flux from the mass balance of the reference layer:

$$b_{15} = b_x - (\overline{L_{15,x}}\rho_{15,x} + F_{15,x}). \tag{5}$$

$\overline{L_{15,x}}$ in equation (5) was simply determined using the recorded core data. We assumed that $\overline{L_{15,x}}$ remained constant over the entire observation period. It is certainly questionable whether this assumption is reasonable as will be discussed later. However, from measured TWT and $\overline{L_{15,x}}$, we could then calculate $\rho_{15,x}$ during periods with dry firn. The third term in equation (5), $F_{15,x}$ corresponds to the gravitational liquid water content of $L_{15,x}$, which can easily be converted from $\theta_w$ if the imaged radar volume is known. We used the same approach as described by Heilig et al. (2015). To assess the imaged radar volume for this layer,

we applied the known radiation characteristics of the radar system. Refraction occurring at density transitions was neglected, since permittivity differences are small and consequently refraction ineffectual. However, for each event with percolating water





reaching $L_{15,x}$, the three phase mixing formula is underdetermined (c.f., Heilig et al., 2015). Hence, to solve for $\theta_w$, we used the same assumption as Heilig et al. (2015) that $\theta_i$ remains constant after initial percolation into $L_{15,x}$. This precondition will be discussed in the following as well.

## 2.4 Regional Climate Model MAR

Here, we use the versions 3.7 and 3.8 of the regional climate model MAR, especially developed for simulating the GrIS surface mass balance. MARv3.7 is run at a resolution of 20 km and is forced by reanalysis NCEP1 (National Centers for Environmental Prediction, resolution of 2.5°) over 1948–2016. MARv3.8 is run at a resolution of 15 km and forced by reanalysis ERA-Interim (ECMWF Interim Re-Analysis, resolution of approximately 0.75°) over 1979–2016. Both reanalyses and the MAR model are described in detail in Fettweis et al. (2017). In respect to MARv3.5 used in Fettweis et al. (2017), the main improvements of MARv3.7 and MARv3.8 - apart from regular bugs corrections - are the increase in cloud life, correcting partly the cloudiness underrepresentation (and, hence, the infrared energy flux) as well as the excess of inland precipitation found in Fettweis et al. (2017). The differences between MARv3.7 and MARv3.8 are mainly improvements in computing efficiency without significant modifications in the physics. The MAR snow model is based on an older version of the snow model Crocus (Brun et al., 1989) using the "bucket approach" as water transport scheme discussed in D'Amboise et al. (2017).

## 3 Results

For the remaining part of this study, we will consistently use "height above the radar antennas" as coordinates for specific horizons and events. All MAR outputs for depths beneath the surface and recorded temperature data are converted to match the radar data. This was performed by subtracting simulated depths beneath the surface from bulk layer thickness of $L_s$ measured by the FirnCover ultrasonic transducer (MacFerrin et al., 2015).

### 3.1 Radar reflection response and corresponding firn core data

All major density steps and ice lenses identified in the cores can be related to radar reflection events (Figure 3a and b). Starting from the bottom, each ice lens corresponds to an amplitude increase in the radargrams. Since we buried the top of the antenna box within the significant ice crust at 0.1 m height, only the decrease in density of that crust produced a reflection response (Figure 3b). The next ice lens at 0.8 m height produced a strong reflection for both frequencies, while the double lens right above at 1.0 m results only in a significant signal amplitude increase in the 1600 MHz radargram (Figure 3a). In firn, the vertical resolution of the 600 MHz antennas is roughly 17.7 cm and for the 1600 MHz antennas 6.6 cm, respectively (Daniels, 2004). Destructive interferences diminish reflections appearing within shorter distance than the respective wavelength. However, the lens at 1.3 m appears again in both radargrams as a strong reflection. This reflection is marked as reference horizon. At about 2.0 m height, we identified another significant ice lens with densities exceeding 800 kg/m$^3$. While for the 1600 MHz array (Figure 3a), it is possible to track this horizon over the entire time period in the radargram, the reflection signal disappears in the 600 MHz data after the last liquid water percolations by mid August (Figure 3b).



The summer 2015 melt produced a remarkable double crust just below the recent snow accumulation at about 2.3 m above the antennas. Both radargrams in Figure 3 show a clear reflection signal for this horizon. The 600 MHz data allow to follow this reflection throughout the whole summer season until fall 2016 (Figure 3b).

Whenever the calculated height of the snow and firn column deviates from the surface reflection signal (e.g. after 19 July, Figure 3b), the used velocity model becomes incorrect due to liquid water infiltrations. Liquid water in snow and firn decelerates radar wave propagation significantly and, consequently, distance to reflections above the infiltration increase in measured TWT. In contrast to the ice lens at about 2 m height, the 2015 horizon and the snow surface, all layers below the reference layer (Figure 3a and b) are basically unaffected by melt events and consequently do not show variations in TWT.

Concerning the surface reflection, different behavior for both antennas could be observed as well. The 1600 MHz radargram (Figure 3a) is incapable of producing a clear surface signal after strong melt affected the snowpack. In contrast, the 600 MHz data still show a clear surface signature. Such occurrences are in agreement with upGPR radargrams observed in seasonal snow (Schmid et al., 2014). The use of a dual-frequency system is beneficial for such events. We still received a strong surface signal even after mid July for the 600 MHz array (Figure 3b).

### 3.2 Validation of radar derived parameters

The calculated layer thickness of the snow and firn column above the antennas $\overline{L_s}$ was compared with data from two ultrasonic depth rangers. One of the ultrasonic transducers is located in a distance of about 60 m to the upGPR location being part of the FirnCover station (MacFerrin et al., 2015) and the other ultrasonic data were measured about 1 km west at the GCnet station (Steffen et al., 1996). Figure 4 displays all three curves. In perennial firn ultrasonic depth rangers measure only the distance of the instruments to the snow surface. Since no snow free conditions can be used to recalibrate the measurements, we adjusted both stations to match the height of the snow and firn column during installation for the start of upGPR measurements. Differences in between ultrasonic data and upGPR determined $\overline{L_s}$ equal to 5.1 cm in comparison to GCnet data and 4.3 cm to results of the FirnCover station in root mean square deviation (RMS) over almost six months of observations.

Density values determined by radar could only be validated through available firn cores, which were drilled during time of visits. Table 1 displays density differences of core data and radar derived values for several different radar reflections, which could be attributed to distinctive layers in cores. As a third data set of validation, we can use the height of the target above the snow surface ($d_A$, Figure 1). In May 2016 this height was measured manually to 1.80–1.86 m, due to surface roughness. In May 2017, we had to raise the target and measured $d_A$ to 2.69–2.70 m. Radar determined $d_A$ equals to 1.79 m in 2016 and 2.68 m in 2017 for the same date as the manual measurements.

### 3.3 Observed temporal changes in snow and firn

In Figure 5, determined changes in snow and firn from upGPR (Figure 5a) are compared with temperature data derived from the installed thermistor string (Figure 5b) and obtained bulk liquid water content (blue curve) and total SWE values (brown curve; Figure 5c). The radargram was additionally processed by horizontal filtering. All reflectors remaining constant over the observation period were thus removed. Such filtering enhances visibility of abrupt changes in stratigraphy such as provoked by



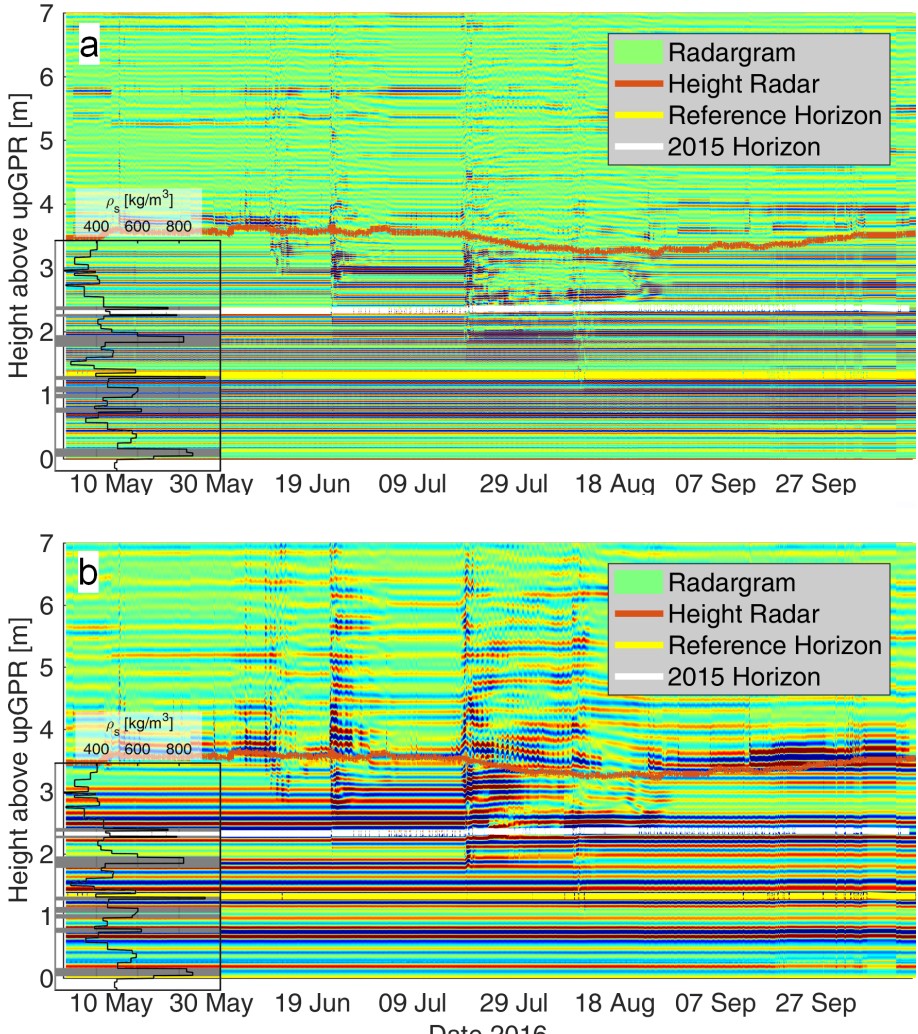

**Figure 3.** Comparison of dual frequency upGPR data with firn core records gathered for the beginning of May 2016. (a) Reflection responses for the 1600 MHz are compared with density and stratigraphy from one firn core with corresponding depth scale. (b) Reflection responses for the 600 MHz are compared with density and stratigraphy from one firn core with corresponding depth scale. Occurrences of ice lenses at specific depths are indicated through gray shaded horizontal areas within the boxes. In addition, we display the determined height of the snow- and firnpack above the antennas (brown line), the height of the reference horizon (yellow line) and the reflection response corresponding to the summer surface of the previous summer (2015 - white line).

water percolation (Figure 5a). In Figure 5b, temperature data are interpolated for the upper four thermistors with the blue line on top indicating the snow surface. Isotherms for every 1 K are presented as black lines.

For the bulk snow and firn above the antennas, we observed two early peaks in melt in June causing percolation to reach down to 2.9 m height in early June and down to 1.8 m on 23 June. After a period of refreezing conditions from early July until





**Table 1.** Measured and radar determined densities for specific layers above the radar antennas. For comparison with core densities, we use the arithmetic mean of both cores.

| Layer | Radar [kg/m$^3$] | Core 1 [kg/m$^3$] | Core 2 [kg/m$^3$] | Deviation to cores [%] |
|---|---|---|---|---|
| Bulk radar 2016 | 479.8 | 472.3 | 495.2 | -1.0 |
| Reference layer 2016 | 449.9 | 436.9 | 468.0 | -1.0 |
| 2015/16 accumulation | 408.8 | 389.9 | 393.2 | +4.2 |
| Bulk radar 2017 | 495.7 | 474.8 | — | +4.4 |
| Reference layer 2017 | 481.4 | 448.3 | — | +7.4 |
| 2015 Summer surface | 452.5 | 417.4 | — | +8.4 |

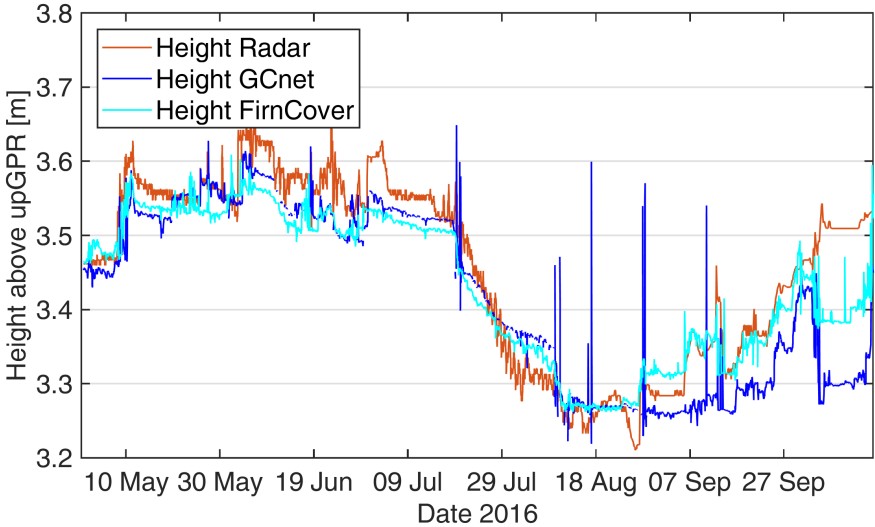

**Figure 4.** Comparison of derived thickness from radar of the snow and firn column above the antennas with changes in snow depth recorded by two different ultrasonic rangers.

mid July, the strong melt event on 19 July caused deep percolation to a height of approximately 1.5 m with derived bulk $\theta_w$ to approach 1 vol%. Melt conditions outlasted until early August when the next increase in melt caused the determined $\theta_w$ to exceed 1.0 vol% and water percolation to reach about 1 m above the antennas. After this peak, we observed rapid refreezing with fully refrozen snow and firn by early September.

5    Table 2 illustrates dates of local minimum for percolation above the radar antennas determined from the radargram and height above the antennas of the −1° C isotherm. This isotherm was determined by linearly interpolating recorded snow temperatures. We decided to interpolate with a resolution of 5 cm in between temperature measurements. In general, radar-observed perco-





**Table 2.** Dates and minimum infiltration heights above the antennas for local minima in percolation of both, the upGPR data and thermistor records. We used the interpolated -1°C isotherm for percolation minima of the thermistor data.

| percolation event | upGPR | thermistor data |
|---|---|---|
| event 1 | 12 Jun. 19:30h 2.9 m | – |
| event 2 | 23 Jun. 01:00h 1.8 m | 27 Jun. 11:00h 3.1 m |
| event 3 | 19 Jul. 17:00h 1.3 m | 19 Jul. 15:00h 2.2 m |
| event 4 | 10 Aug. 21:00h 1.0 m | 10 Aug. 19:00h 0.9 m |

lation matches well the temperature progression. Almost all liquid water occurrences in the radar data at the snow surface or below (indicated in the radargram by distinct multiples or ringing above the surface up to 7 m in air) correspond to heat waves penetrating into deeper layers of snow and firn. While the first stronger melt event by early June did not affect temperature records significantly, the next melt event for this season showed a clear signal in temperature data as well. The delay in tem-

perature response by more than four days in Table 2 is a consequence of the simple search for local minimum in height of the −1°C isotherm. The primary decrease in height of that isotherm occurred already on 23 June at 21:00 h and consequently was delayed only by 20 h in comparison to upGPR results. However, the minimum height within the melt period of the isotherm was reached four days later. The strongest dips in water percolation for mid July and early August 2016 match by 2 h for radar and thermistor string. The measurements of percolation depths differ more significantly. The first two percolation events

recorded by temperature data mismatch radar observed percolation depths by roughly 1 m. However, for the strongest event by early August the coincidence of radar and temperature observations is 10 cm. Actual temperature records for the same day showed a maximum of -0.21°C at a height of 1.0 m at 17:00 h (Figure 5b). The given accuracy of the deployed thermistors is in the range of ±0.25 °C. Even though the minimum in height of percolation for the radar was detected four hours later (Table 2), we detected percolation reaching a height of approximately 1.1 m in the radar data at 17:30 h the same day. Concerning

determined $\theta_w$ data in Figure 5c, it occurs that any strong gradient in derived $\theta_w$ correspond well with timing of percolation of the warming signal for the temperature records.

Since all contributing volume fractions of the overlying snow- and firnpack are determined, we can simply calculate for accumulation mass in water equivalent as well. The bulk SWE over the antennas is presented in Figure 5c (brown line). During wet snow conditions the determined SWE remained stable or only slightly increased. Just after 01 September and before 01

June remarkable increases in accumulation were determined.

## 3.4 Seasonal Mass Transfer

We could clearly identify a strong mass transfer from contemporary snow into firn below the 2015 summer surface (Figure 6a and b). At least three melt events routed liquid water beneath this summer horizon (Figures 3a, b, 5a and b), which was located at about 2.4 m above the antennas for May 2016. In total, we determined a mass flux of 56.4 kg/m$^2$ from snow into

firn (Figure 6b). The three major percolation events occurred after mid June and before mid August. While the first event





**Figure 5.** (a) Radargram of the observed six month time period in 2016 with display of water percolation. (b) Recorded and interpolated snow and firn temperatures with 1 °C contour interval for the upper four thermistors of the installed thermistor chain. The cyan line in (b) represents the snow surface measured by the FirnCover ultrasonic transducer. (c) Derived bulk volumetric liquid water contents above the antennas (blue line, left axis) in comparison to radar data of changes in total mass in snow water equivalent (SWE) for the same layer (brown line, right axis). The dashed lines in (c) represent the uncertainty of SWE arising from the error in density and layer thickness determinations.

produced an outflow of roughly 6 kg/m² water mass from the snow layer in three individual routing events within three hours, the percolations in July and August routed 27 kg/m² and 23.5 kg/m² respectively (Figure 6b). $L_{15,x}$ experienced a volumetric liquid water content of up to almost 1 vol% at 10 August 2016 18:00 h (Figure 6a, blue line). At that day, both the thermistor





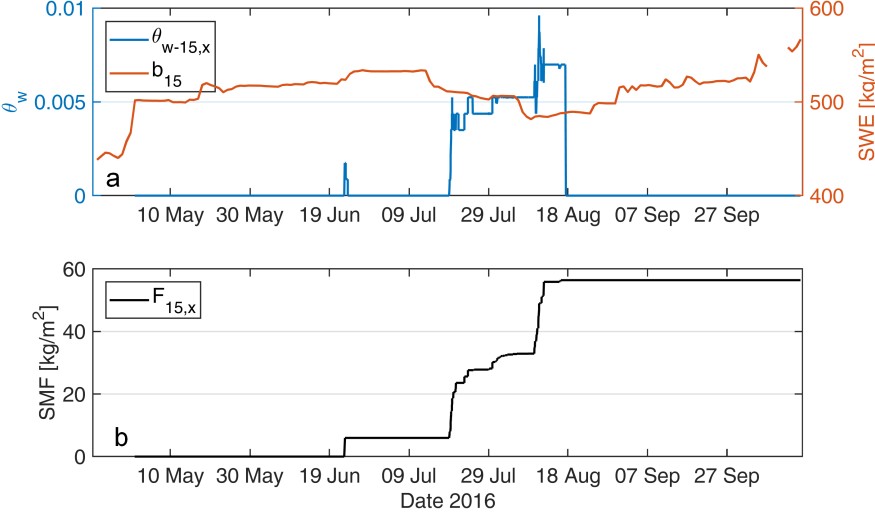

**Figure 6.** (a) Mass balance estimates for the snow layer above the summer horizon 2015 (brown line, right axis) and changes in $\theta_w$ of the layer between the summer horizon of 2015 and the reference horizon (blue line, left axis). (b) Seasonal mass flux (SMF) that has percolated through the 2015 summer horizon into firn below.

data and radar observations obtain the maximum depth in percolation (Table 2). The timing of all three data sets is within three hours difference.

The mass balance estimates for early May 2016 ($b_{15}$) derived from the radar exceeds conventionally measured SWE values for the snow layer by roughly 100 kg/m² (upGPR $b_{15} = 438.2$ kg/m²; $b_{15}$ measured in the pit above the antennas: 334.8 kg/m²).

This difference can be attributed to difficulties in picking the seasonal snow above the summer horizon of 2015 in the radargram. The reflection response at this specific density gradient is masked by signal interferences with the reflection generated at the lower border of the ice lens of summer 2015. Correspondingly, including the observed ice lenses into SWE calculation of the pits results in an mass of 426.0 mm w.e. for early May. This reduces the offset to values obtained from upGPR to only 2.8%.

### 3.5    Comparison of radar derived snow parameters with simulations from MAR

MAR is forced every 6 h by either NCEP1 or ERA-Interim reanalysis data. These forcings generate MAR outputs with a daily temporal resolution and grid cells of 20 km (NCEP1) and 15 km (ERA-Interim), respectively. The radar, in contrast, provided point data on changes in total accumulation of up to every 30 min together with data on volumetric liquid water content, percolation and bulk density (Figures 5, 6).

The comparison of seasonal changes in accumulation in between simulation results and radar data (Figure 7) shows high

agreement for both data sets. Here, we averaged radar data to diurnal outputs to match resolution of MAR results. Uncertainty in radar determined SWE derives from the error in total height of snow ($\pm 4.3$ cm, Section 3.2) and the uncertainty in density estimates ($\pm 1\%$, Tab. 1) in an error propagation. Apart from the beginning of the time series in May, changes in SWE simulated



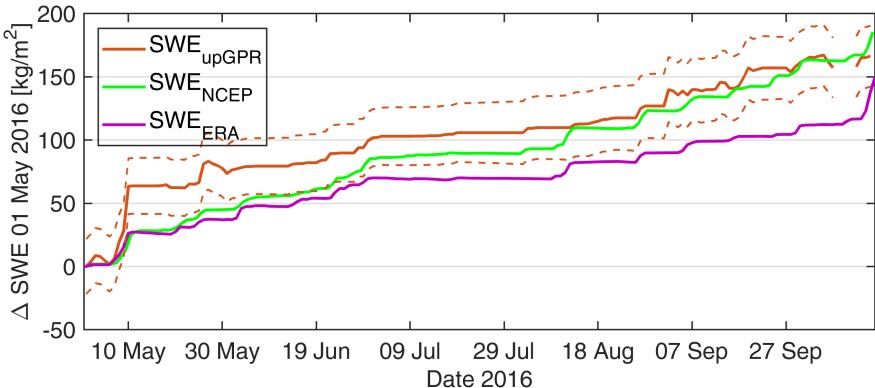

**Figure 7.** Seasonal changes in accumulation in respect to 01 May 2016. We compare upGPR derived changes in SWE (brown line with uncertainty range indicated by dashed lines) with simulated variations by MAR for both forcings (green line - NCEP1 forcing; purple line - ERA-Interim forcing).

in MAR with both forcings match radar observations very accurately. To assess the similarity between simulations and radar data, we calculate the Nash-Sutcliffe efficiency value (NSE) (Nash and Sutcliffe, 1970), which is at 0.75 for MAR simulations using NCEP1 forcing and below 0 for ERA-Interim driven simulations for the whole data series. While NCEP1 driven simulations gradually approach changes determined from upGPR data over time, MAR with ERA-Interim forcing remain parallel

to the radar line almost over the entire time series. We assume that the strong rise in SWE for upGPR data at 10 and 11 May 2016 is attributed to additional drifting caused by a shelter, which we created for digging the radar pit. Hence, removal of the strong increase in SWE occurring at 11 May ($\Delta b_s = 35.4 \, \mathrm{kg/m^2}$) for radar data lead to NSE values for MAR-NCEP1 of 0.53 and MAR-ERA of 0.95. Consequently, the temporal progression of changes in SWE is simulated in MAR with very high agreement to radar data using ERA-Interim forcings and acceptably well with NCEP1 forcing (Figure 7). However, the

simulated significant increase in accumulation (by MAR-NCEP1) at 10 August is not reproducible by radar observations and distinctly smaller for ERA-Interim forced MAR.

For $\theta_w$ and bulk snow density above the reference horizon much more distinct differences in between simulations and radar determinations appear (Fig. 8a and b). Bulk density over the entire observation period is highly overestimated by MARv3.7 with NCEP1 forcing and significantly underestimated by MARv3.8 with ERA-Interim forcing for this specific location. Bulk

density values of the NCEP1 forced simulation are exaggerating field data within the full observation period. While simulations overestimate $\rho_x$ in the beginning by only $20 \, \mathrm{kg/m^3}$, at the peak of the melt season, differences of almost $100 \, \mathrm{kg/m^3}$ are commonly present (Figure 8a). RMS deviations to upGPR derived $\rho_x$ for MAR forced by NCEP1 reach $71.4 \, \mathrm{kg/m^3}$. RMS values determined for ERA-Interim forced MAR simulations result in $51.2 \, \mathrm{kg/m^3}$, which is only slightly better and still represents a deviation of roughly 10% in comparison to mean $\rho_x$. Here, MAR models bulk density of the upper 2 m constantly too low.

MAR simulations with both forcings tend to exaggerate melt at Dye-2. This is especially the case for MAR being forced by NCEP1 reanalysis. For instance, the first spike in simulated $\theta_w$ for mid May does not have an equivalent in radar data at all.



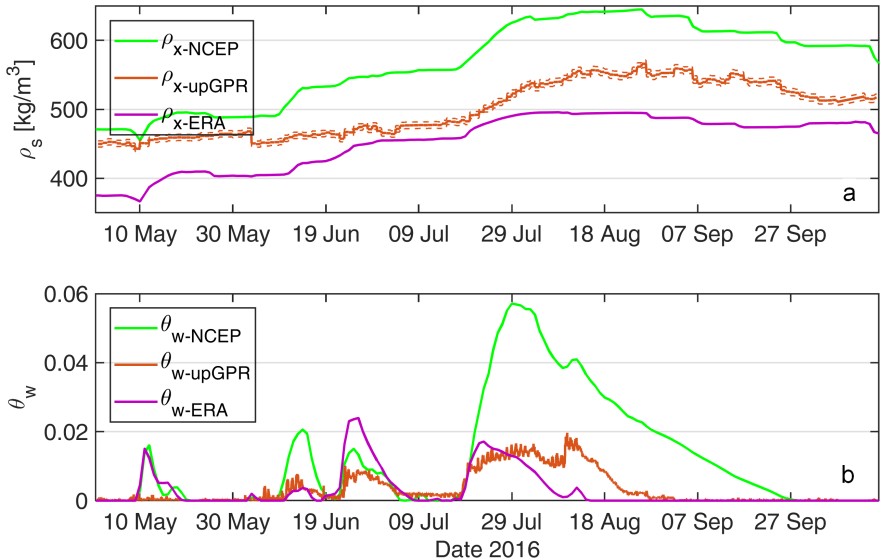

**Figure 8.** (a) Seasonal changes in bulk density ($\rho_x$) for layer $L_x$ simulated by MAR with NCEP1 and ERA-Interim forcing compared with $\rho_x$ derived from upGPR data (brown line with uncertainty range). (b) Seasonal changes in $\theta_w$ for the same layer $L_x$ simulated by MAR with forcing NCEP1 and ERA-Interim compared with $\theta_w$ values from radar data (brown line). For bulk density in (a) as well as bulk liquid water content in (b) upGPR data has a temporal resolution of 30 min maximum, while MAR has daily values as output.

Here, MAR simulations exaggerate the amount of melt and the duration. Documented changes in snow temperature (Figure 5b) do not indicate such strong melt occurrences either. The subsequent simulated $\theta_w$ peaks correspond in timing but not in amplitude for MAR-NCEP1, while ERA-Interim forced MAR matches the amplitude but refreezes earlier. For the melting period lasting from 23 June until 3 July timing of the melt event agrees with radar derived data. Here, ERA-Interim forcing

leads to a stronger overestimation in amplitude than NCEP1. Such occurrences are opposite for the subsequent melt event starting at 19 July. While MAR-ERA data agree well in $\theta_w$ amplitude with radar, MAR-NCEP1 overestimates maximum $\theta_w$ by almost a factor of three. In consequence, refreezing is delayed for MAR-NCEP1 by 27 days. Since MAR-ERA misses the strong peak in melt (10 August), refreezing is simulated already for 15 August 2016 and thereby 18 days earlier than radar data indicates (Figure 5b). Temporal offsets in between diurnal average values of $\theta_w \geq 0.3\,\mathrm{vol\%}$ for the upGPR and

NCEP1 forced simulations are always within maximum one day for the initiation of melt. However, duration of the periods with $\theta_w \geq 0.3\,\mathrm{vol\%}$ differ by three days in mid June and 31 days in late August/ September 2016. For MAR-ERA onset of melt reaching $\theta_w = 0.3\,\mathrm{vol\%}$ is delayed by three days in mid June and otherwise within $\pm 1$ day. Refreezing of snow and firn to values below $0.3\,\mathrm{vol\%}$ is usually predicted within an accuracy of $\pm 1$ day as well with exception of mid August, when MAR-ERA simulates a drop below the $0.3\,\mathrm{vol\%}$ range 15 days too early.

Simulations of percolation depths for both model forcings are highly diverse and mainly disagree with upGPR determined data (Figure 9). Temporal agreement for the onset of melt is high for MAR-NCEP1 and upGPR percolation but percolation



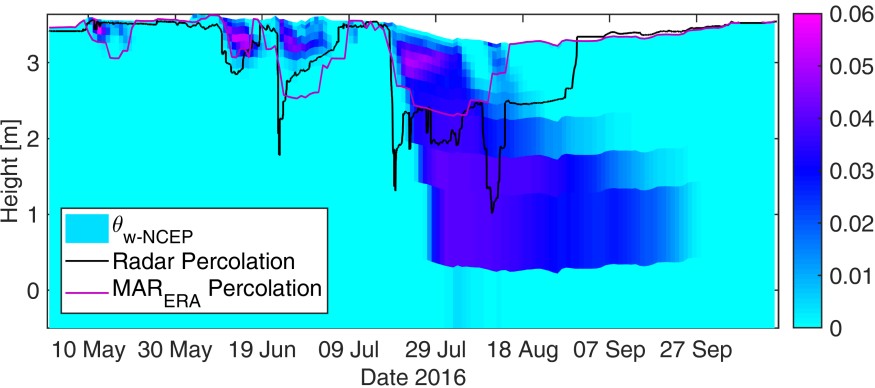

**Figure 9.** Simulated layer specific $\theta_w$ with depth compared with upGPR derived percolation depths (black line). The color bar presents volumetric liquid water content from MAR-NCEP1.

depths and timing of refreezing do not agree. For MAR-ERA, percolation depths are mostly underestimated over the course of the season and the strong melt in August is not captured, which leads to an earliness of refreezing. Both percolation simulations exceed radar determined percolations significantly for the first melt event in mid May, which is in agreement with bulk $\theta_w$ predictions. For the following melt occurrences at mid June, offsets in maximum percolation are rather small. Radar data

reveal a height of infiltrating liquid water down to 2.85 m above the antennas, MAR-NCEP1 down to 2.82 m and MAR-ERA down to 3.07 m. Here, MAR-ERA has a slight delay in timing of water infiltration. The following melt event lasting from late June to early July results in much larger offsets of percolation depths. Deviations to radar data are at -1.08 m (MAR-NCEP1) and -0.76 m (MAR-ERA). For the major melt event (19 July – mid August), MAR-NCEP1 exceeds maximum percolation as observed by radar by +0.81 m and MAR-ERA underestimates water infiltration by -1.27 m. Such percolation offsets are in

agreement with $\theta_w$ over- (MAR-NCEP1) and underestimation (MAR-ERA) as shown in Figure 8. For both simulations, the speed of percolation is significantly underestimated for the onset of melt, when compared with upGPR data.

For the time period in between 03 August until 08 August, we observed refreezing conditions at the bottom of the percolation (Figure 5a, 9). MAR-NCEP1, however, simulates a stable percolation front with refreezing being simulated at the snow surface (Figure 9). ERA-Interim forced simulation correctly predicts refreezing from the bottom.

# 4 Discussion

## 4.1 Reliability of radar derived snow- and firnpack parameters

It is important to mention that the velocity model used for assessment of heights above the antennas is less accurate after liquid water entering the snow and firnpack. Here, percolating water decreases the wave velocity, which increases TWT for layers being situated above the infiltration. However, snow pits and firn cores at the site can only be obtained when the instruments are





visited once a year. For data analysis only measured TWT is used and, consequently, presented heights are not relevant. Still we consider a presentation of heights, even though they are partly incorrect after certain time periods, as being more intuitive and more supportive for readability. Percolation depths are unaffected from erroneous TWT conversions since they indicate the maximum height of dry snow and firn.

In Section 2.2, we described four assumptions required to enable data derivation for wet snow conditions: (i) no lateral flow transported mass downslope; (ii) wind erosion after surficial wetting is negligible; (iii) evaporation and sublimation effects are negligible for wet snow surfaces as well and (iv) no mass loss above the antennas is possible as long as percolation did not reach antenna height. Assumption (i) and (iv) induce each other and, hence, are discussed together. Lv et al. (2013) conclude that lateral redistribution of soil moisture is sensitive to slope angle. Here, we observed an area with an almost plain surface ($< 0.5°$

slope angle). Consequently, lateral redistribution of liquid mass is considered negligible. Considering liquid water percolation, we recorded changes in firnpack stratigraphy every 30 min during daytime. For none of the records water infiltration past the radar antennas was identifiable. There is a slight chance that small amounts of water percolated in between two radar measurements below the depth of the antennas and refroze before the next radar scan. Such infiltration, however, would cause a release of latent heat at such depth during refreezing, which is not documented in the temperature data (Figure 5b). Wind

erosion of wet surfaces is assumed to have a negligible effect, since cohesion forces and bonds among grains are much stronger than for loose new snow (Li and Pomeroy, 1997). For the proof of assumption (iii), we used MAR outputs and quantified the effect of sublimation and evaporation during melting surfaces. For the time period in between 19 July and 19 August 2016, when strong melt affected the snow and firn at Dye-2 (Figure 5), MAR calculates an effect of evaporation being at 5% of simulated SMB. Such an effect remains within the given uncertainty for radar derived SWE.

Due to the fact that independent snow and firn temperature records of $T \geq -1°\,$C match percolation observed by radar very accurately and due to the high agreement between seasonal changes in SWE simulated with MAR and radar determined SWE development, we have strong reasons to trust results derived from radar data. In addition, calculated $\overline{L_s}$ above the antennas is in close agreement with two time series of ultrasonic depth rangers. An error of 4–5 cm ($< 1.5\%$ for a 3.4 m thick snow- and firnpack) is below an observed uncertainty between manual measurements and snow depth sensors for a much smaller spatial

offset in seasonal snow (Schmid et al., 2014). For the presented data, conventionally measured bulk densities for specific layers agreed within $\pm 1\%$ with radar derived densities for May 2016. In 2016, we had the opportunity to drill cores less than 2 m from the center of the radar antennas. Overestimation of bulk density of radar data in May 2017 cannot be directly attributed to increased uncertainties in radar derived parameters. Due to the fact that we did not want to influence snow and firn within the footprint of the radar antennas, we had to drill the core in 2017 about 8 m away from the center of the target reflector. Spatial

variability in stratigraphy and $\overline{L_s}$ caused difficulties in relating layers to radar reflectors and contributed to offsets for specific layer densities. The height of the target reflector above the snow surface could be determined with very high accuracies as well.

The assumption of a fixed layer thickness in section 2.3 for $L_{15,x}$ bases on the fact that during cold and dry conditions the TWT for both determined horizons remain at the same sample number within $\pm 1$ sample uncertainty. In addition, it is important to consider the respective firn layer to be part of a closed system. Neither evaporation, sublimation nor erosion can transfer

mass. Due to rather small temperature gradients in perennial firn (here, approximately 3 K/m at maximum; Figure 5b), water





vapor transport mechanisms are small and consequently negligible. We presume that only compaction with a corresponding increase in $\rho_s$ influence the measured TWT for dry conditions. Theoretically, it is possible that compaction is happening but the measured TWT remains constant. For instance, such conditions could be the case for the period until 19 June 2016 (Figure 3). The numerical approximation for a fixed TWT with varying $\rho_s$ values ranging from $200 - 900$ kg/m$^3$ results in

$s = 1.5 \times 10^{-7}\rho_s^2 - 5.1 \times 10^{-4}\rho_s + 1.4$, with the strain $s$ in meter and $\rho_s$ in kg/m$^3$. From this approximation it follows that a density increase for the observed layer of $\Delta\rho_s = +100$ kg/m$^3$ would only allow a strain of about 3.7 cm for the reflector remaining at the same distance in TWT. For an observation period of one year, we observed maximum density increases of less than 30 kg/m$^3$ per layer (Table 1). Thus, the fixed layer thickness is a reasonable assumption for possible densification rates of that layer.

In addition, we assume the ice volume fraction to remain constant for the time period after water reached the respective layer and before refreezing is completed. Such an assumption is conceptually wrong in cold firn. Percolating water will refreeze and through the release of latent heat gradually increase the temperature of this layer. However, a gradual increase in $\theta_i$ is difficult to estimate from the given temperature resolution of the thermistor data. Consequently, we overestimate $\theta_w$ after initial percolation. However, only further increases in $\theta_w$ result in further increases in the amount of $F_{15,x}$ within the layer.

Since $F_{15,x}$ remains stable after the first percolation event reaching $L_{15,x}$ (23 June) and after the third event (10 August), we expect the named overestimation to being of relevance only for the period in between 19 July and 10 August. In consequence, for this time period of gradual warming (see Figure 5b), the assumption of $\theta_i = const$ might lead to an overestimation of less than 10 kg/m$^2$ for $F_{15,x}$.

## 4.2   Changes in seasonal snow and firn for the melt season 2016

For the summer season 2016, we observed several major changes in snow and firn parameters. According to the radar records, a maximum volumetric liquid water content of $\theta_w \leq 2$ vol% was observed for snow and firn above the reference layer (approximately 2 m beneath the snow surface). A maximum percolation depth throughout the season of 1.0 m height above the antennas, which corresponds to 2.3 m below the surface was recorded for 10 August. Deep percolation down to 10 m and more as proposed by Machguth et al. (2016) for the here observed elevation range could not be observed for the melt season in 2016.

In terms of spatial extent of melt at the surface, this melt season is considered as above average (tenth in the 38-year satellite records) (NSIDC, 2016). All melt events together routed about 60 kg/m$^2$ of mass into firn beneath the previous summer surface of 2015. This corresponds to roughly 40% of liquid water, which were transferred into deeper layers, while about 60% were retained against gravitational forces within the seasonal snow layer. Steger et al. (2017a) model an average retention over the entire GrIS of 47% with values reaching up to 75% in the south-east of Greenland where rates of snow accumulation are

largest. We could not observe major stratigraphic changes along the previous summer surface after the melt season 2016 as proposed by Pfeffer and Humphrey (1998); neither within the radargrams of both frequencies nor in the firn core of May 2017. However, a distinct increase in accumulation for the layer above the reference horizon and below summer 2015 was recorded from May 2016 ($b_{15,x} = 483.7$ kg/m$^2$) to May 2017 ($b_{15,x} = 533.9$ kg/m$^2$) of $\Delta b_{15,x} = 50.2$ kg/m$^2$. This confirms the recorded mass transfer, despite of radar determined mass transfer being ~12% larger. Spatial inhomogeneities and inaccuracies in both



measurement methods (uncertainty through use of $\theta_i = const$, difficulties in layer attribution within firn cores) certainly contribute to this offset. Although, one should be very cautious of direct comparisons between annual firn cores, especially for individual layers, a general trend of mass increase could be confirmed by this core data. However, it is obvious that small scale changes appeared within the course of the melting period in 2016. In the layer bonded by the summer 2015 and the reference

horizon, remarkable changes in reflection structure occur after percolation. Especially, the 600 MHz signal was influenced. A new reflector appeared right below the summer 2015 horizon and the reflection previously attributed to the significant ice lens at about 2 m height diminished with refreezing firn.

Concerning the mass balance of the snow layer above the summer horizon 2015 ($b_{15}$) at Dye-2, we found an increase in accumulation of 84.4 kg/m$^2$ for the time period of May until 30 September 2016. The simulated SMB in MAR resulted

in 151.3 kg/m$^2$ for the same time span with a simulated mass loss of only 6.8 kg/m$^2$. Subtracting the mass flux of $F_{15,x} = 56.4$ kg/m$^2$ of mass would result in an overestimation in MAR of $b_{15}$ in comparison with radar data of roughly 12%. This is in agreement with results presented by e.g., Heilig et al. (2015) that model accuracies benefit from in situ data. For assessment of mass balance rates at Dye-2 without runoff and lateral redistributions at the current stage, it is of no relevance whether mass is transferred into firn beneath or remains within the seasonal accumulation layer. Concerning lower elevation sites at the

transition between accumulation and ablation area, however, the accurate assessment of residual water and outflow is critical for estimates on mass balances (Charalampidis et al., 2015). The same appears for the formation of near surface layers of low permeability (Machguth et al., 2016). Only monitoring and accurate determination of liquid mass being transferred into firn enables correct simulation of ice layer formations and future development.

### 4.3 Reliability of model simulations in comparison with upGPR data

Generally, regional climate model outputs are not compared with data from single point measurements and validation on time spans of days to several months is not common (Fettweis et al., 2017). It remains questionable whether such comparisons are fruitful or not, keeping in mind that the modeled snowpack is representing a mean state over an area of $20 \times 20$ km$^2$ ($15 \times 15$ km$^2$). However, since conventional instrumentations such as lysimeters to measure snowpack outflow or snow pillows to determine changes in SWE are not applicable in perennial firn, upGPR offers an unique possibility to validate - on a temporal

continuous basis - simulated snow and firn parameters with measurements and determine reliability of model results. Hence, we tested the performance of MAR on its upper end of accuracy.

In general, the performance of MAR with both forcings is very good especially for the timing of melt onset and simulated changes in SWE. After removal of one data point supposedly influenced by drifting, the agreement of seasonal SWE changes of upGPR data and simulations reach up to 0.95 in NSE values for the ERA-Interim forcing. Such NSE values indicate an almost

perfect fit of simulation data. The temporal offset of melt simulated by MAR with respect to upGPR and thermistor results is mostly below one day, which is the temporal resolution of the model outputs. Such accurate performance of a regional climate model is encouraging since the model is not run with input data from the AWS nearby but forced at its lateral boundaries with atmospheric fields with a typical resolution of 100 km. As a consequence, the downsampling of MAR seems to be reasonably



accurate. It should be remembered that we compare point measurements of specific parameters with an average snowpack over $20 \times 20\,\text{km}^2$ ($15 \times 15\,\text{km}^2$) in area, which likely partially explains discrepancies.

Significant offsets between simulations and radar observations exist for the calculation of bulk density of the upper 2 m in snow and firn, which reach an offset of up to $+100\,\text{kg/m}^3$. In addition $\theta_w$ is overestimated for each melt event up to a factor

of three in comparison to values derived for the upGPR. The general exaggeration of melt in the percolation zone by regional climate models has been described previously for another model as well (Noël et al., 2015). As a consequence of overestimation of density and $\theta_w$ for MAR run by NCEP1 forcing, water percolates too deep and refreezing is strongly delayed. The irreducible liquid water content of snow and firn is related to porosity (Schneider and Jansson, 2004). Snow and firn of higher density have less potential to retain liquid water and thus percolation is overreached. MAR forced by ERA-Interim, however, has a tendency

to exaggerate bulk volumetric liquid water content as well but with a lower amplitude. For two out of four melt events during the summer 2016, MAR-ERA predictions of $\theta_w$ are in agreement with radar data over a few days. However, MAR forced by ERA-Interim misses the peak of melt and percolation in August 2016 almost completely. For the moment when upGPR data obtain the highest percolation depths, MAR-ERA simulates refreezing in snow and firn. Here, problems with the reanalysis forcing might occur, which lead to a distinct underrepresentation of melt. Simulation of liquid water infiltration and percolation

depths are coupled with the amount of melt being produced at the surface and the applied water transport scheme. The here used simple bucket approach is not capable in reproducing water infiltration as observed by radar and temperature data. Deviations of simulation results for percolation depths are rather erratic. This surveillance is in agreement with previous comparisons in seasonal snow (e.g., Wever et al., 2015). The bucket approach is not capable in predicting heterogeneous infiltration and consequently, percolation is delayed at each onset of strong melt events but once melt has started is routing liquid water too

fast in deeper snow (Wever et al., 2015). This study displays a very similar behavior of the bucket scheme for perennial firn as well. However, in contrast to seasonal snow, the cold content in firn forces refreezing from the bottom of water percolation as long as latent heat release is absorbed by the cold content of the surrounding firn. Hence, the typical water infiltration pattern of sharp dips in height as documented by radar and temperature data (i.e. Figure 5) is not reproduced in the model independent of used forcings. In addition, without adequate climate forcing, melt cannot be predicted in a correct manner. Neither of the

two applied forcings for MAR enable correct prediction of full snowpack refreezing. Hence, we conclude that a model capable in modeling heterogeneous flow is required to assess water infiltration, retention and refreezing correctly.

## 5   Conclusions

This study investigated temporal changes of liquid water content, density and SWE in snow and the upper few meters of perennial firn within the deep percolation zone of the GrIS. Over the entire melt season in 2016, liquid water infiltrations reach

a minimum height above the radar antennas of 1 m, which corresponds to 2.26 m beneath the snow surface. The volumetric liquid water content does not exceed 2 vol% for the upper approximate 2 m beneath the snow surface. It is obvious from radar data that liquid mass has been routed out of the snow layer into firn beneath. We obtain a seasonal mass flux of $56.4\,\text{kg/m}^2$





for the six months observation period in 2016. The applied instrumentation enable quasi-continuous monitoring of changes in mass for specific layers as well. For the bulk layer above the antennas, we derive a change in mass of $+156.6\,\mathrm{kg/m^2}$.

We compare results derived from upGPR data with MAR run by two different reanalysis forcings and modeling a mean snow- and firnpack over an area of $20 \times 20\,\mathrm{km^2}$ ($15 \times 15\,\mathrm{km^2}$ respectively). In general, the performance of MAR with both forcings is very good especially for the timing of melt onset and simulated temporal changes in SWE. However, prediction of layer density and bulk liquid water content is inaccurate for both reanalysis. ERA-Interim forced MAR is slightly decreasing the offset in density and significantly improving the performance for simulation of bulk $\theta_w$. This study demonstrates that for correct assessment of infiltration depths and timing of refreezing, a more sophisticated water transport scheme than the bucket approach is required.

On a long-term perspective the installation of upGPR antennas at such a location might provide observation data on the transition from porous firn into either the formation of impermeable ice slabs or the gradual filling of the pore space above. Since the spatial melt extent in 2016 over the GrIS derived from remote sensing data was among the ten largest of the last 38 years, we do not expect percolation to reach beneath the height of the antennas apart from very exceptional years such as 2012. This possibly will enable monitoring of melt, mass fluxes and accumulation at this site for the next years to come.

*Data availability.* All parameters derived from upGPR data are available from the lead author upon request together with raw radar data. The MAR outputs are freely available under ftp://ftp.climato.be/fettweis/MARv3.8/

*Competing interests.* The authors declare that they have no conflict of interest

*Acknowledgements.* A. Heilig was supported by DFG grant (HE 7501/1-1). Computational resources for running MAR have been provided by the Consortium des Équipements de Calcul Intensif (CÉCI), funded by the Fonds de la Recherche Scientifique de Belgique (F.R.S.–FNRS) under grant no. 2.5020.11 and the Tier-1 supercomputer (Zenobe) of the Fédération Wallonie-Bruxelles infrastructure funded by the Walloon Region under the grant agreement no. 1117545. M. MacFerrin was supported by the National Aeronautics and Space Administration (NASA) grant NNX15AC62G. M. Tedesco would like to acknowledge NSF Award PLR # #1604058 and NASA Award #NNX17AH04G. We highly acknowledge support in logistics and preparation of the field campaigns from K. Young and staff from Polar Field Services. In addition, we would like to thank S. Williams and D. Abbott for swapping and mailing SD cards at the end of each summer. L. Schmid and M. Siebers supported software preparations and T. Sponholtz helped with instrument preparations. Field assistance by B. Gerling, L. Gambal, S. Samimi, S. Marshall, M. Stevens, B. Vandecrux, J. Kingslake, C. Miege and F. Covi was highly acknowledged.



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
