# Peer review of "Seasonal monitoring of melt and accumulation within the deep percolation zone of the Greenland Ice Sheet and comparison with simulations of regional climate modeling"

_The Cryosphere, 2017_

## Referee Comment (RC1) · Anonymous Referee #1 · 27 Jan 2018

Review of

Seasonal monitoring of melt and accumulation within the deep percolation zone of the Greenland Ice Sheet and comparison with simulations of regional climate modeling

by Heilig and others

Summary

This paper addresses an important topic, the observation of snow mass, liquid water content and percolation depth in the firn layer of the Greenland deep percolation zone.

[Figure]

The method has been used before in seasonal snow packs and uses upward looking radar, which is less destructive than similar studies using thermistors and other sensors that must be inserted in the snow. A comparison is then made with output of a regional atmospheric climate model (MAR). Good agreement is found in the magnitude and timing of snow accumulation events, but important biases are identified for bulk snow density, liquid water content, percolation depth and refreezing rate and depth.

Assessment

The paper is original with good scientific quality and impact. The technical quality of the manuscript leaves some things to be desired; the text must be cleaned up, the introduction section restructured, see comments below. These are all relatively minor issues.

Major comments

Unfortunately, the manuscript text contains multiple typo's and generally suffers from mediocre readability because of frequent unclear formulations. For instance, the introduction requires close scrutiny, several statements are unclear or inaccurate, see (non-exhaustive) listing below under minor comments. The introduction would also benefit from a more logical structure starting with the mass loss from the Greenland ice sheet and then stepwise building up to the importance of meltwater retention, its observation (especially the lack thereof) and model evaluation.

Adjust the number of significant digits throughout the manuscript to represent the real accuracy of the results. For instance (both on page 12), temperature probably is less significant than stated at 0.01 degree, and mass transfer and/or SWE not at 0.1 kg m-2.

Section 3.5: Surely, MAR output must be available at a higher time resolution than 1 day? The model time step must be typically several minutes, so the line 10 statement that "these generate MAR output with a daily temporal resolution" appears inaccurate.

Why does forcing MAR with different reanalysis data (NCEP and ERA-Interim) produces such large differences in surface density and refreezing characteristics, if the same snow model is used (Figs. 8 and 9)? This touches directly upon model performance and deserves to be discussed in more detail. A paragraph with discussion of possible causes for the biases found in the model is also warranted.

Minor comments

p. 1, l. 5: are observable -> have been observed (?). Also p. 2, l. 27 and 28.

p. 2, l. 4: "...average negative mass balance all over the ice sheet...": this is unclear; what you probably mean to say is that the ice sheet mass balance became persistently negative.

p. 2, l. 5: multiplied -> increased

p. 2, l. 6: "...Negative annual surface mass balances over the same time period are attributed to an increase in surface melt and runoff...": inaccurate: surface mass balance integrated over the ice sheet has not yet been negative; locally it has, of course. Do you mean: Negative trends in surface mass balance?

p. 2, l. 9: "...Since melt conditions are expected to continue...": unclear; do you mean: are expected to continue to increase?

p. 2, l. 16: "...cause a large fraction of uncertainty cause a large fraction of uncertainty...": consider replacing with: "...is a major component of the uncertainty..."

p. 3, l. 1: "...few existing automatic weather stations ...": Nowadays there are two major WS networkd on the GrIS: GC-Net and PROMICE.

Table 2 does not add much information, and its contents can be absorbed in the main text.

p. 12, l. 20: "...increases in accumulation ...": increases in SWE (?)

p. 12, l 22: how is 'contemporary snow' defined?

---

## Referee Comment (RC2) · Anonymous Referee #2 · 5 Mar 2018

**1 Overview/Summary:**

This manuscript describes the deployment of an upward-looking ground-penetrating radar (upGPR) in the Greenland firn. Using the upGPR, the authors observe percolation of water into the snowpack, and movement of water within the firn, measuring percolation depths and deriving paramaters such as bulk density and volumetric water fraction. The authors compare the measurements of these percolation parameters with the outputs of a popular regional climate model, MAR, as forced by two different reanalysis datasets. In general this is an interesting manuscript that advances knowlege in

the field, and is worthy of publication. The science is good with no serious flaws. The presentation is a bit lacking, primarily in the use of English. I've commented extensively (though probalby not exhaustively) in the 'minor comments' section below.

2 Major Comments

- Page 3, Lines 16-20... Seems to me like you can either use MAR to help determine how the radar does, or use the radar to determine how MAR does, but not both. At all.

- Page 4, Lines 31-33... Not convinced that the 'compaction of the radar and the target' are actually equal; in practice the target supports may be driven down faster than the radar. Effect is likely small but will be non-zero.

- In general, much of the Results section appears to be more like Discussion; I have not attempted here to disentangle the discussion-like parts from the straight results, but this is a part of the paper that could use refinement.

3 Minor Comments

- Abstract Line 2: 'act as meltwater' -> 'act as a meltwater'

- Abstract Line 9: 'capable to monitor quasi-continuously' -> 'capable of continuously monitoring' (though use quasi if you must)

- Page 2, Line 5: 'multiplied by a factor' -> 'has increased by a factor'

- Page 2, Lines 7-8: sentence needs rewording; should start '61% of the recent mass loss is ascribed to...'

- Page 2, Line 19: 'are exposed' -> 'have been exposed'

- Page 2, Line 23: Reference to Abdalati and Steffen 2001- this is a remote-sensing paper, not looking at in-situ data.

- Page 2, Line 25: unless McFerrin et al is published by the time this comes to press, remove this reference.

- Page 2, Line 29 'inevitable'- is this the right word? Sounds like you meant to say 'imperative'.

- Page 3, Line 2: 'temporal continuous' -> 'temporally continuous'

- Page 5, Line 9: 'identification of timing' -> 'identification and timing'

- Page 5, Line 11: reference to figure 3 before I see any reference to figure 2, the first reference I see to which is on page 6.

- Page 6, Line 5: Seems 120 kg/mˆ3 is pretty low for Greenland snow (to me) but I don't have a reference to point to.

- Page 8, Line 26: delete 'respectively' as it's not needed here.

- Page 9, Line 4: I don't understand what the first part of this sentence is referencing; I don't see the change after 19 July that I think the statement is discussing.

- Page 9, Line 21: 'equal to' -> 'are'

- Page 9, Lines 30-31: This description needs to be tightened up. The 'determined changes in snow and firn' are really 'extent of percolation'; the results you show in panel c are also 'changes in snow and firn'...

- Figure 5b: Do you really believe the high-frequency variability of the isotherms (most prominent example is between 30 may and 19 Jun going from 1 to 0 m height)? I don't. These should probably be smoother curves. Filtering these data might be what you need to do before calculating isotherms. The other really major thing here is that because you discuss the -1 C isotherm so much in the manuscript, it should be delineated clearly here- a different color, or marked in some way. This way readers can see the trends you are describing on page 12 in time and depth of the -1 C isotherm.

- Page 11, Line 2: 'outlasted' -> 'lasted'

- Page 12, Line 9: Start a new paragraph here.

- Page 12, line 23: this assertion would be easier to verify if you plotted the 2015 summer surface on figure 5; that would show clearly the melt propagating below the summer 2015 suface (harder to verify quickly by ooking at figures 6a and b as referenced).

- Page 14, Lines 5-8: Not sure why all of this is here. Why wouldn't you include the ice lenses that you observed in the pits? In that case simply report the last sentence on line 8. But it's not clear if these ice layers were measured in the pits or where they used from the radar to 'correct' the pit data? If so, not sure why it's valid to do that. Clarity of the language is required here.

- Page 15, Line 2: Rather than force the reader to look up the physical meaning of the Nash-Sutcliffe efficiency value (which one needs in order to evaluate your reported numbers), briefly define it here. Only needs a sentence.

- Page 15, Line 8: I think this kind of correction is probably fine- if you want to quote the statistics for it, you need to demonstrate how "removal of the strong increase" was done, and it would probably be good to show the resultant curve on figure 7.

- Page 17, Figure 9: I don't understand why the theta_w NCEP is illustrated the way it is. Seems like on previous figures there was a parallel track with MAR-NCEP and MAR-ERA, and this should continue in this figure.

- Page 17, Line 18: 'entering' -> 'enters'

- Page 18, Line 9: 'plain' -> 'planar'

- Page 18, Line 16: I'm not sure that using MAR provides adequate proof of assumtion iii. Isn't MAR modeling these physical processes using assumptions of its own? An observational proof would be more useful here. However, I think iii is a very reasonable assumption and don't think it really needs a rigoroous proof.

- Page 19, Line 6: Strain is by definition a dimensionless quantity, if you are using a measurement of 3.7 cm, this must be deformation, or shortening, not strain.

- Page 19, Line 24: 'could not be' -> 'was not'

- Page 19, Line 30: 'could not' -> 'did not'

---

## Author Comment (AC1) · 28 Mar 2018

We thank both referees for very constructive and helpful comments. Each comment of each referee is considered separately in the following. In addition, minor changes such as typos were incorporated in the MS without listing them here. In order to improve readability, comments by the respective referee are listed in italic, while responses and modifications in the MS are written regularly.

Please also note the supplement to this comment:

[Figure]

https://www.the-cryosphere-discuss.net/tc-2017-277/tc-2017-277-AC1-supplement.pdf

[Figure]

**Supplement:**

We thank both referees for very constructive and helpful comments. Each comment of each referee is considered separately in the following. In addition, minor changes such as typos were incorporated in the MS without listing them here. In order to improve readability, comments by the respective referee are listed in italic, while responses and modifications in the MS are written regularly.

**Reply to referee #1:**

We highly appreciate comments raised by the referee and present a point-to-point reply for all issues raised by the referee. For an improved readability and to facilitate direct response, we sometimes subdivided comments into several paragraphs referring to similar issues

*Comments to the Author*
*Assessment*
*The paper is original with good scientific quality and impact. The technical quality of the manuscript leaves some things to be desired; the text must be cleaned up, the introduction section restructured, see comments below. These are all relatively minor issues.*
We appreciate the assessment by the reviewer.

*Major comments*
*Unfortunately, the manuscript text contains multiple typo's and generally suffers from mediocre readability because of frequent unclear formulations. For instance, the introduction requires close scrutiny, several statements are unclear or inaccurate, see (non-exhaustive) listing below under minor comments. The introduction would also benefit from a more logical structure starting with the mass loss from the Greenland ice sheet and then stepwise building up to the importance of meltwater retention, its observation (especially the lack thereof) and model evaluation.*
We adjusted the structure of the introduction in accordance to the suggestions made by the reviewer. To improve readability (see also comments raised by Rev#2), we carefully revised larger parts of the manuscript and checked thoroughly for typos. We sincerely apologize, if this appeared to be too sloppy before initial submission.

*Adjust the number of significant digits throughout the manuscript to represent the real accuracy of the results. For instance (both on page 12), temperature probably is less significant than stated at 0.01 degree, and mass transfer and/or SWE not at 0.1 kg m-2.*
We agree with the reviewer and adjusted given accuracies. However, since the accuracy of the temperature sensors is stated by the manufacturer to 0.25 K, we would like to keep this statement in the manuscript.

*Section 3.5: Surely, MAR output must be available at a higher time resolution than 1 day? The model time step must be typically several minutes, so the line 10 statement that "these generate MAR output with a daily temporal resolution" appears inaccurate.*
You are correct, MAR outputs are available with sub-daily temporal resolution. However, it remains debatable, whether sub-daily outputs for a 20 km x 20 km grid are valuable for comparison with sub-daily upGPR measurements for a single point. As stated in the manuscript (section 4.3), we still test MAR on its upper end of accuracy. We came to the conclusion that daily averages are adequate for such comparison. However, we changed the respective statements. "We use MAR outputs with a daily temporal resolution and two different forcings, which generate grid cells of 20 km (NCEP1) and 15 km

(ERA-Interim), respectively." In addition, we modified the respective part in the methodology to: "MAR is forced every 6 h by either NCEP1 or ERA-Interim reanalysis data. We decided to use daily outputs for comparisons."

*Why does forcing MAR with different reanalysis data (NCEP and ERA-Interim) produces such large differences in surface density and refreezing characteristics, if the same snow model is used (Figs. 8 and 9)? This touches directly upon model performance and deserves to be discussed in more detail. A paragraph with discussion of possible causes for the biases found in the model is also warranted.*

We included the following paragraph into the discussion section: "As stated above, predicting individual parameters of the SMB for a point location of the GrIS is beyond the scope of regional climate modeling. Here, we used two different versions of MAR with two different resolutions. This explains already a large fraction of the observed discrepancies for the analyzed parameters density and melt. Since models are usually tuned to accurately reproduce SMB data, individual parameters such as bulk density or bulk liquid water content may result in variable offsets from in-situ data for different climate forcings. In addition, the initial conditions for summer 2016 for both ERA-Interim and NCEP1 are not exactly equal, which causes the model to adjust differently for the individual parameters. Next, clouds have a large impact on the energy balance of the percolation zone of the GrIS. Due to the positive feedback of melt and albedo, small differences in the timing of melt and the amount result in significant offsets for the used forcings. However, upGPR data can help to identify misconceptions in regional climate modeling and, consequently, support further improvements in simulations of temporal changes in snow- and firnpacks."

*Minor comments:*
*p. 1, l. 5: are observable -> have been observed (?). Also p. 2, l. 27 and 28.*
Changed accordingly

*p. 2, l. 4: "...average negative mass balance all over the ice sheet...": this is unclear; what you probably mean to say is that the ice sheet mass balance became persistently negative.*
Changed to "… which resulted in persistent negative mass balances all over the ice sheet"…

*p. 2, l. 5: multiplied -> increased*
Changed accordingly

*p. 2, l. 6: "...Negative annual surface mass balances over the same time period are attributed to an increase in surface melt and runoff...": inaccurate: surface mass balance integrated over the ice sheet has not yet been negative; locally it has, of course. Do you mean: Negative trends in surface mass balance?*
Changed accordingly to negative trends

*p. 2, l. 9: "...Since melt conditions are expected to continue...": unclear; do you mean: are expected to continue to increase?*
Changed accordingly

*p. 2, l. 16: "...cause a large fraction of uncertainty cause a large fraction of uncertainty...": consider replacing with: "...is a major component of the uncertainty..."*
Replaced accordingly

*p. 3, l. 1: "...few existing automatic weather stations ...": Nowadays there are two major AWS networkd on the GrIS: GC-Net and PROMICE.*

Changed to: "Apart from several existing automatic weather stations (AWS) being unevenly distributed over the GrIS, no temporal continuous observations exist to validate results of such models."

*Table 2 does not add much information, and its contents can be absorbed in the main text.*
Here, we respectfully disagree. You are correct that the content could easily be absorbed in the main text. However, this would require most likely the same space as the table. We consider having a table as much more supportive for such simple dates and numbers than several sentences describing this.

*p. 12, l. 20: "...increases in accumulation ...": increases in SWE (?)*
Changed to: …snow accumulation…

*p. 12, l 22: how is 'contemporary snow' defined?*
Contemporary has been deleted.

---

## Author Comment (AC2) · 28 Mar 2018

We thank both referees for very constructive and helpful comments. Each comment of each referee is considered separately in the following. In addition, minor changes such as typos were incorporated in the MS without listing them here. In order to improve readability, comments by the respective referee are listed in italic, while responses and modifications in the MS are written regularly. Listed page and line numbers refer to the previously submitted manuscript.

[Figure]

Please also note the supplement to this comment:
https://www.the-cryosphere-discuss.net/tc-2017-277/tc-2017-277-AC2-supplement.pdf

**Supplement:**

**Reply to referee #2:**

We highly appreciate comments raised by the referee and present a point-to-point reply for all issues raised by the referee. For an improved readability and to facilitate direct response, we sometimes subdivided comments into several paragraphs referring to similar issues

*This manuscript describes the deployment of an upward-looking ground-penetrating radar (upGPR) in the Greenland firn. Using the upGPR, the authors observe percolation of water into the snowpack, and movement of water within the firn, measuring percolation depths and deriving paramaters such as bulk density and volumetric water fraction. The authors compare the measurements of these percolation parameters with the outputs of a popular regional climate model, MAR, as forced by two different reanalysis datasets. In general this is an interesting manuscript that advances knowlege in the field, and is worthy of publication. The science is good with no serious flaws. The presentation is a bit lacking, primarily in the use of English. I've commented extensively (though probalby not exhaustively) in the 'minor comments' section below.*

We thank the referee for the evaluation and the positive assessment.

*2 Major Comments*

*- Page 3, Lines 16-20... Seems to me like you can either use MAR to help determine how the radar does, or use the radar to determine how MAR does, but not both. At all.*

This is correct and we apologize for the previous version; now changed to "To estimate the reliability of radar-derived parameters, we compare determined percolation depths with changes in temperature records and analyze monitored changes in thickness of the snow and firn column above the antennas with results of ultrasonic transducers located within a distance of less than 2 km (Steffen et al., 1996; MacFerrin et al., 2015)."

*- Page 4, Lines 31-33... Not convinced that the 'compaction of the radar and the target' are actually equal; in practice the target supports may be driven down faster than the radar. Effect is likely small but will be non-zero.*

Here, we respectfully disagree that this is the case given the current level of accuracy. Such an unequal compaction would most likely be gradually and would have effects on the radar derived snow/ firn height data. If the compaction of the posts holding the radar target would be faster, $d_A$ would decrease and since d is taken as being constant $L_s$ would gradually increase. The results should show up as systematic gradual overestimation of the radar derived $L_s$. This is obviously not the case in Fig. 4. However, in the manuscript, we state "to be approximately equal".

*- In general, much of the Results section appears to be more like Discussion; I have not attempted here to disentangle the discussion-like parts from the straight results, but this is a part of the paper that could use refinement.*

We sincerely apologize for the sloppiness and checked carefully for appearances of discussions in the Results section. Among other parts previous P9 L4-6, P12 L5-8 as well as P14 L5-8 are now moved to the discussion part or changed to meet requirements for a results section. Please see the respective marked parts in the discussion section.

*3 Minor Comments*
*- Abstract Line 2: 'act as meltwater' -> 'act as a meltwater'*

*- Abstract Line 9: 'capable to monitor quasi-continuously' -> 'capable of continuously monitoring' (though use quasi if you must)*
*- Page 2, Line 5: 'multiplied by a factor' -> 'has increased by a factor'*
All have been changed accordingly.

*- Page 2, Lines 7-8: sentence needs rewording; should start '61% of the recent mass loss is ascribed to...'*
The sentence has been revised to:" van den Broeke et al. (2016) attributed 61% of the recent mass loss to a decrease in surface mass balances and only 39% to an increase in solid ice discharge."

*- Page 2, Line 19: 'are exposed' -> 'have been exposed'*
Changed

*- Page 2, Line 23: Reference to Abdalati and Steffen 2001- this is a remote-sensing paper, not looking at in-situ data.*
Changed to: "Information on melt usually just exist for the area extent of surficial melt over the GrIS (e.g., Abdalati and Steffen,2001) from remote sensing data."

*- Page 2, Line 25: unless McFerrin et al is published by the time this comes to press, remove this reference.*
We changed the reference to Machguth et al. (2016) and slightly modified to "massive ice lenses"

*- Page 2, Line 29 'inevitable'- is this the right word? Sounds like you meant to say 'imperative'.*
Changed to essential

*- Page 3, Line 2: 'temporal continuous' -> 'temporally continuous'*
*- Page 5, Line 9: 'identification of timing' -> 'identification and timing'*
Both changed accordingly

*- Page 5, Line 11: reference to figure 3 before I see any reference to figure 2, the first reference I see to which is on page 6.*
We switched the order of the figures.

*- Page 6, Line 5: Seems 120 kg/m^3 is pretty low for Greenland snow (to me) but I don't have a reference to point to.*
We dug numerous pits around Dye-2 in May 2016. For conditions with recent new snow, we found density values of 104 kg/m$^3$, 52.8 kg/m$^3$, 93.3 kg/m$^3$ and 61.8 kg/m$^3$ for the upper 5-10cm. The 120 kg/m$^3$ seems to be an appropriate estimate, especially while accounting for the progress of the season including warmer temperatures in summer. We are pretty confident that this value is not too low. We ask the reviewer to specify why the value should be higher at this particular location, e.g. by providing a reference.

*- Page 8, Line 26: delete 'respectively' as it's not needed here.*
Changed accordingly.

*- Page 9, Line 4: I don't understand what the first part of this sentence is referencing; I don't see the change after 19 July that I think the statement is discussing.*
We moved this paragraph into the discussion and modified to:
"The used wave speed model becomes incorrect when liquid water infiltrates snow and firn. Liquid water decelerates radar wave propagation significantly and, consequently, distance to reflections above the infiltration increase in measured TWT. However, snow pits and firn cores at the site can only be obtained

when the instruments are visited once a year. For data analysis only measured TWT is used and, consequently, presented heights are not relevant. Still we consider a presentation of heights, even though they are partly incorrect after certain time periods, as being more intuitive and more supportive for readability. Percolation depths are unaffected from erroneous TWT conversions since they indicate the maximum height of dry snow and firn. In contrast to the ice lens at about 2 m height, the 2015 horizon and the snow surface, all layers below the reference layer (Figure 2a and b) are basically unaffected by melt events and consequently do not show variations in TWT."
We hope this will enhance comprehension for readers, which are not familiar with TWT domains.

*- Page 9, Line 21: 'equal to' -> 'are'*
Changed accordingly

*- Page 9, Lines 30-31: This description needs to be tightened up. The 'determined changes in snow and firn' are really 'extent of percolation'; the results you show in panel c are also 'changes in snow and firn'...*
Modified to: "In Figure 5, determined changes in snow and firn from upGPR (Figure 5a: extent of percolation and c: changes in SWE - brown line and volumetric liquid water content - blue line) are compared with temperature data derived from the installed thermistor string (Figure 5b)."

*- Figure 5b: Do you really believe the high-frequency variability of the isotherms (most prominent example is between 30 may and 19 Jun going from 1 to 0 m height)? I don't. These should probably be smoother curves. Filtering these data might be what you need to do before calculating isotherms. The other really major thing here is that because you discuss the -1 C isotherm so much in the manuscript, it should be delineated clearly here- a different color, or marked in some way. This way readers can see the trends you are describing on page 12 in time and depth of the -1 C isotherm.*
The referee is right that the oscillating signal of the isotherms is actually noise. Here, we are at the stated 0.25 K accuracy of the sensors. Those sensors were deployed in May 2016 and, consequently, required some time to settle and measure the environmental temperature. The noise in the signal is even much more prominent for the very first time period after installation (Fig. 5b). For the revised version, we smoothed the signal. Just for presentation, it is most likely better to present processed data, however, for analysis it is insufficient to smooth/ filter the data. Percolation depths are significantly altered by any kind of processing see figure below. Hence, data in Tab. 1 is derived from recorded raw data.

[Figure]

This figure demonstrates the effect of smoothed thermistor data. We used PT100 temperature sensors. The blue line displays the raw temperature signal interpolated to generate a -1°C isotherm and its location in height above the antennas. The orange line represents the same isotherm, while PT100 data has been smoothed using a quartic Savitzky-Golay filter.

*- Page 11, Line 2: 'outlasted' -> 'lasted'*
Changed accordingly

*- Page 12, Line 9: Start a new paragraph here.*
Changed accordingly.

*- Page 12, line 23: this assertion would be easier to verify if you plotted the 2015 summer surface on figure 5; that would show clearly the melt propagating below the summer 2015 suface (harder to verify quickly by ooking at figures 6a and b as referenced).*
Here, we refer to figures 6a and b, which present quantification of seasonal mass transfers from snow into firn below summer surface 2015. Fig. 5 just present percolation depths. We decided to keep the reference and did not modify Fig. 5. The chosen presentation in Fig. 5a is horizontally filtered, which means that the relatively stable 2015 layer is filtered out. We believe that including a horizontal line in this figure would distract and confuse the reader, since the purpose of this figure is to demonstrate the agreement of thermistor data with radar recordings and derived parameters.

*- Page 14, Lines 5-8: Not sure why all of this is here. Why wouldn't you include the ice lenses that you observed in the pits? In that case simply report the last sentence on line 8. But it's not clear if these ice layers were measured in the pits or where they used from the radar to 'correct' the pit data? If so, not sure why it's valid to do that. Clarity of the language is required here.*

This is another paragraph, which we shifted into the discussion part. We revised it to enhance readability to: "Offsets in radar derived mass balance data ($b_{15}$) of about 100 kg/m$^2$ to manual observations can be attributed to difficulties in picking the reflection event seasonal snow above the summer horizon of 2015 in the radargram. Snow pits are usually just dug down to a remarkable crust, which is hardly penetrable with a shovel. The reflection response at this specific density gradient is masked by signal interferences with the reflection generated at the lower border of this crust, which represents the melt horizon of summer 2015. Correspondingly, including the observed ice lenses into SWE calculation of the pits results in a mass of 426 kg/m$^2$ for early May. This reduces the offset to values obtained from upGPR to only 2.8%."

*- Page 15, Line 2: Rather than force the reader to look up the physical meaning of the Nash-Sutcliffe efficiency value (which one needs in order to evaluate your reported numbers), briefly define it here. Only needs a sentence.*
The sentence has been included: "NSE values of 1 indicate a perfect fit of the model with the data, while a NSE of 0 shows that the model fit is as good as simply the average value of the data. NSE for MAR NCEP1 simulations is at 0.75 and below 0 for ERA-Interim driven simulations for the whole data series."

*- Page 15, Line 8: I think this kind of correction is probably fine- if you want to quote the statistics for it, you need to demonstrate how "removal of the strong increase" was done, and it would probably be good to show the resultant curve on figure 7.*
We agree that the description of this kind of correction was misleading. The respective sentences were modified to: "Hence, deleting only the data point of 11 May ($\Delta bs$ = 35 kg/m$^2$) from analysis lead to NSE values for MAR-NCEP1 of 0.53 and MAR-ERA of 0.95."
So, we simply removed this data point from NSE analysis. It is incorrect to plot an additional SWE curve in Fig. 7. Here, we present cumulative changes in SWE in respect to 1 May conditions. Certainly, snow accumulation happened for the 11 May. It just might not have been as large as measured by the radar due to the disturbed wind field from the shelter causing drifting snow at the position of the upGPR. However, to analyze similarity of two curves by NSE values, it is appropriate to remove single data points, if they are biased.

*- Page 17, Figure 9: I don't understand why the theta_w NCEP is illustrated the way it is. Seems like on previous figures there was a parallel track with MAR-NCEP and MAR-ERA, and this should continue in this figure.*
We decided to plot simulated liquid water content per specific layers for NCEP1 to demonstrate offsets between model runs and data. Such presentation provides more information than just a height curve with simulated θw=0. Referee #1 explicitly asked for a more detailed description and explanation of differences in between both forcings. We consider such presentation as being supportive for this. However, we detected an error in the previous interpolation scheme of the figure plot. We changed the figure to a contour plot with contour intervals at 1 vol%. Due to the coarse depth resolution of MAR below 2 m from the surface (next data point at 3 m depth), we decided to plot the interpolated depth of the 0.1 vol% as approximate for the bottom of the wetting front to address the recommendation of the referee.

*- Page 17, Line 18: 'entering' -> 'enters'*
*- Page 18, Line 9: 'plain' -> 'planar'*
Changed accordingly

*- Page 18, Line 16: I'm not sure that using MAR provides adequate proof of assumption iii. Isn't MAR modeling these physical processes using assumptions of its own? An observational proof would be more*

*useful here. However, I think iii is a very reasonable assumption and don't think it really needs a rigoroous proof.*

We added the following sentences to describe limitations of this proof: "However, MAR uses assumptions as well to estimate sublimation and evaporation. According to our knowledge, no experimental setup within the deep percolation zone of the GrIS exists to provide a more rigorous proof for assumption (iii)."

*- Page 19, Line 6: Strain is by definition a dimensionless quantity, if you are using a measurement of 3.7 cm, this must be deformation, or shortening, not strain.*

Changed to compaction.

*- Page 19, Line 24: 'could not be' -> 'was not'*
*- Page 19, Line 30: 'could not' -> 'did not'*

Changed accordingly